# FG-CIBGC: A Unified Framework for Fine-Grained and Class-Incremental Behavior Graph Classification

## Abstract

Learning-based Behavior Graph Classification (BGC) has been widely adopted in Internet infrastructure for partitioning and identifying similar behavior graphs. However, the research communities realize significant limitations when deploying existing proposals in real-world scenarios. The challenges are mainly concerned with (i) fine-grained emerging behavior graphs, and (ii) incremental model adaptations. To tackle these problems, we propose to (i) mine semantics in multi-source logs using Large Language Models (LLMs) under In-Context Learning (ICL), and (ii) bridge the gap between Out-Of-Distribution (OOD) detection and class-incremental graph learning. Based on the above core ideas, we develop the first unified framework termed as **F**ine-**G**rained and **C**lass-**I**ncremental **B**ehavior **G**raph **C**lassification (**FG-CIBGC**). It consists of two novel modules, i.e., gPartition and gAdapt, that are used for partitioning fine-grained graphs and performing unknown class detection and adaptation, respectively. To validate the efficacy of FG-CIBGC, we introduce a new benchmark, comprising a new 4,992-graph, 32-class dataset generated from 8 attack scenarios, as well as a novel Edge Intersection over Union (EIoU) metric for evaluation. Extensive experiments demonstrate FG-CIBGC's superior performance on fine-grained and class-incremental BGC tasks, as well as its ability to generate fine-grained behavior graphs that facilitate downstream tasks. The code and dataset are available at: https://anonymous.4open.science/r/FG-CIBGC-70BC/README.md.

## CCS Concepts

• **Mathematics of computing** → **Graph algorithms**; • **Networks** → **Network security**.

## Keywords

Fine-grained Behavior Graph Classification, Class-Incremental Graph Learning

## 1 Introduction

Sophisticated attacks increasingly threaten the global internet infrastructure. As graph offers an ideal representation for security investigation [13], analysts often transform audit logs into a large, unified graph containing numerous operations. However, navigating and investigating the large-scale graph presents a non-trivial challenge of heavy analysis workload [16]. Behavior Graph Classification (BGC) addresses this challenge by partitioning the large graph into a set of smaller behavior graphs and subsequently classifying them, enabling analysts to focus on a few representative behaviors. BGC has emerged as an indispensable technique for various security investigation domains [47], including Host Intrusion Detection Systems (HIDSs), vulnerability detection, *etc*.

Existing solutions on BGC task can be categorized into three types: pattern-based [48, 54], rule-based [12, 14, 15, 30], and learning-based [42, 46]. The former two rely on static patterns and expert

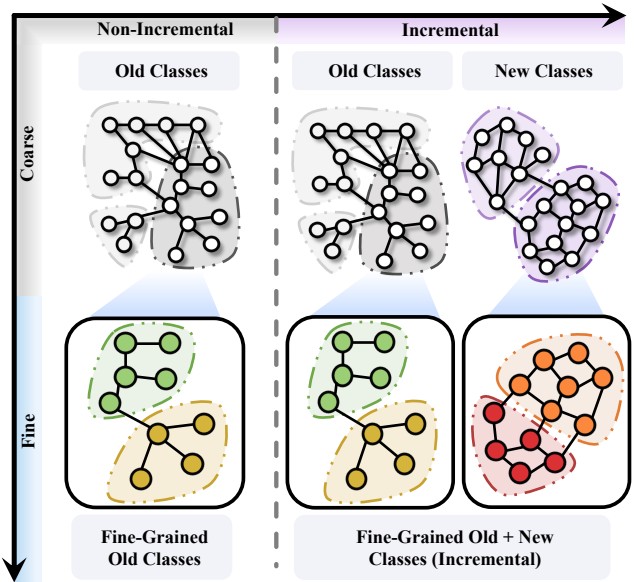

**Figure 1: An illustration of fine-grained and class-incremental behavior graph classification task. This task faces both *fine-grained emerging behavior graphs* and *incremental model adaptations* challenges, leading to performance degradation of state-of-the-art baselines.**

knowledge, demanding heavy manual effort. Learning-based methods have addressed this limitation by leveraging machine learning models. While it sounds promising, the research communities have uncovered a series of limitations when implementing the learning-based approaches in real-world scenarios. By summarizing those issues in Fig. 1, we recognize the following two main challenges.

**(i) Fine-Grained Emerging Behavior Graphs.** Prior learning-based proposals rely solely on coarse-grained behavior graphs. That is to say, while these methods can ascertain whether a behavior graph is related to a specific service (e.g., "apache"), they lack the granularity to differentiate between distinct operations of that service, such as distinguishing "apache processing request 1" from "apache processing request 2". Partitioning discrete operations from large graphs generated by audit logs remains a formidable challenge, as existing approaches consolidate all operatons on a given object into a single graph, limiting their ability to differentiate distinct service operations. Yet fine-grained labels are pivotal for analysts to comprehend service activities and deploy effective countermeasures. Existing coarse-grained BGC approaches present a significant *semantic gap* between graph identification and actionable intelligence. A more granular scheme capable of automatically distinguishing distinct service operations would substantially enhance the understanding of attack vectors and facilitate targeted

defenses. *Consequently, the primary challenge lies in developing a methodology for fine-grained behavior graph partition.*

**(ii) Incremental Model Adaptations.** In real-world scenarios, behavior graphs evolve in an incremental manner, presenting the requirement of *class increments*. *Class increments* refer to emerging novel classes that should be incrementally updated into the model to become known classes subsequently. In production environments, continuous service updates introduce novel behavior graph classes unknown to analysts (also known as the *open-world* issue). Actively detecting and attaching new classes to a model's knowledge base without *catastrophic forgetting* is a significant challenge. Despite the importance, graph-level class-incremental learning with novel class detection remains a largely unexplored area.

In this paper, we propose the first unified framework called **F**ine-**G**rained and **C**lass-**I**ncremental **B**ehavior **G**raph **C**lassification (**FG-CIBGC**), aiming to enable two vital abilities, i.e., the fine-grained behavior graph partitioning and incremental model updates with novel class detection. At the high level, FG-CIBGC is designed with two novel components named *gPartition* and *gAdapt*. **First**, gPartition processes multi-source logs by leveraging Large Language Models (LLM) under In-Context Learning (ICL) paradigm to correlate semantically similar logs, forming *behavior units*. Each *behavior unit* corresponds to a single operation performed by a service application and is subsequently converted into a compact behavior graph. **Second**, gAdapt is responsible for utilizing Out-Of-Distribution (OOD) detection to identify unknown classes, assigns fine-grained labels to both known and unknown classes of behavior graphs, and incrementally updates the model accordingly. **Finally**, existing BGC benchmarks suffer from insufficiencies in both completeness (lacking multi-source logs) and diversity, as well as the absence of metrics tailored to the fine-grained requirements of BGC tasks. To address these issues, we introduce a new benchmark comprising 8 attack scenarios and 3 log types, amounting to 4,992 graphs across 32 classes. Additionally, we propose a novel metric, Edge Intersection over Union (EIoU), to fully evaluate FG-CIBGC.

In summary, this paper makes three key contributions:

- Through analysis of current learning-based behavior graph classification proposals, we identify two critical challenges that impede their deployment. To tackle these challenges, we propose FG-CIBGC, the first unified framework for fine-grained and class-incremental behavior graph classification.
- We design two novel components (i.e. gPartition and gAdapt) for FG-CIBGC, thereby realizing the fine-grained behavior graph identification and incremental model update with novel class detection simultaneously.
- We construct a new benchmark that satisfies both *completeness* and *diversity*, featuring 3 log types, 4,992 graphs across 32 classes, as well as a new EIoU metric for fine-grained evaluation. Extensive experiments demonstrate the superiority of FG-CIBGC.

## 2 RELATED WORK

**Behavior Graph Classification.** Existing BGC methods can be divided into three categories: **(i) pattern-based methods** [48, 54] mine graph patterns from behaviors of interest and use them as templates to identify similar behaviors; **(ii) rule-based methods** [14,

15, 30, 31] match audit events against a knowledge store of rules that describe behaviors; **(iii) learning-based methods** [42, 46] utilize machine learning models to represent behavior graphs as vectors, enabling identifying of semantically similar behaviors. Compared with prior work, we pioneer the exploration of class-incremental BGC task, thereby demonstrating greater competence in real-world scenarios. Furthermore, we are the first to produce fine-grained behavior graphs matching operations in services.

**LLMs-based Log Processing.** In recent years, with the increase in model sizes and richer training corpora, LLMs have notably grown in power. Given the vast pretraining datasets that encompass logging-related data, LLMs possess immense potential for log processing tasks. Existing research has explored the application of large language models across a wide range of log-related tasks, including **log parsing** [18, 39, 40, 51], **vulnerability detection** [35, 37] and **anomaly detection** [9, 26, 32]. Compared with prior works, we uniquely explore the capabilities of large language models in correlating multi-source log data. Furthermore, we have designed a novel type-position-aware prompt format to enable more effective in-context inference of log correlation using the LLMs.

**Class-Incremental Graph Learning.** Recently, class-incremental graph learning has garnered growing attention owing to its broad applications [7, 41], with existing works in this domain generally falling into three primary categories: **regularization-based** [22], **architecture-based** [49], and **replay-based** [34, 52] methods. Crucially, existing methods have largely assumed that all data comes from a predefined set of classes known to humans. However, this assumption does not hold for behavior graph classification task in real-world scenarios, where previously unknown classes may emerge during the learning process. Compared with prior works, we are the first to bridge the gap between Out-of-Distribution (OOD) detection methods and class-incremental graph learning, thereby handling *class increments* in real-world scenarios.

## 3 Problem Definition

Our goal is to incrementally identify and classify semantically similar behavior graphs within a stream of multi-source logs. Given that fine-grained and class-incremental behavior graph classification involves two critical challenges, we provide precise definitions for each of these respective aspects.

**Fine-Grained Emerging Behavior Graphs.** Given a prior dataset $\mathcal{D}_{tra}$ comprising multi-source logs (e.g., audit, application, and network logs), we first extract multiple fine-grained behavior graphs $\mathcal{G}_{tra}$ from the dataset. The term *fine-grained behavior graph* denotes a representation where each behavior graph precisely specifies an operation of a service. Each behavior graph $G_i \in \mathcal{D}_{tra}$ corresponds a category label $y_i \in \mathcal{Y}_{tra}$, where $\mathcal{Y}_{tra} = \{y_{tra}^1, y_{tra}^2, \cdots, y_{tra}^n\}$ where $n$ refers to the number of known classes. And we use $\mathcal{D}_{tra}$ as the training set to fit the model $\mathcal{M}$. When deploying the model $\mathcal{M}$ in practice, it will encounter the open-world test set $\mathcal{D}_t$ at stage $t$, which includes: (i) samples whose ground-truth labels are present in the training set $\mathcal{D}_{tra}$; and (ii) instances of emerging unknown classes $\{y_t^1, y_t^2, \cdots, y_t^m\}$, where $m$ denotes the number of unknown classes, which is unknown to us a priori.

**Incremental Model Adaptations.** The *class increments* represent an inherent challenge in class-incremental learning, and it distinguishes our approach from existing techniques. We assume there is

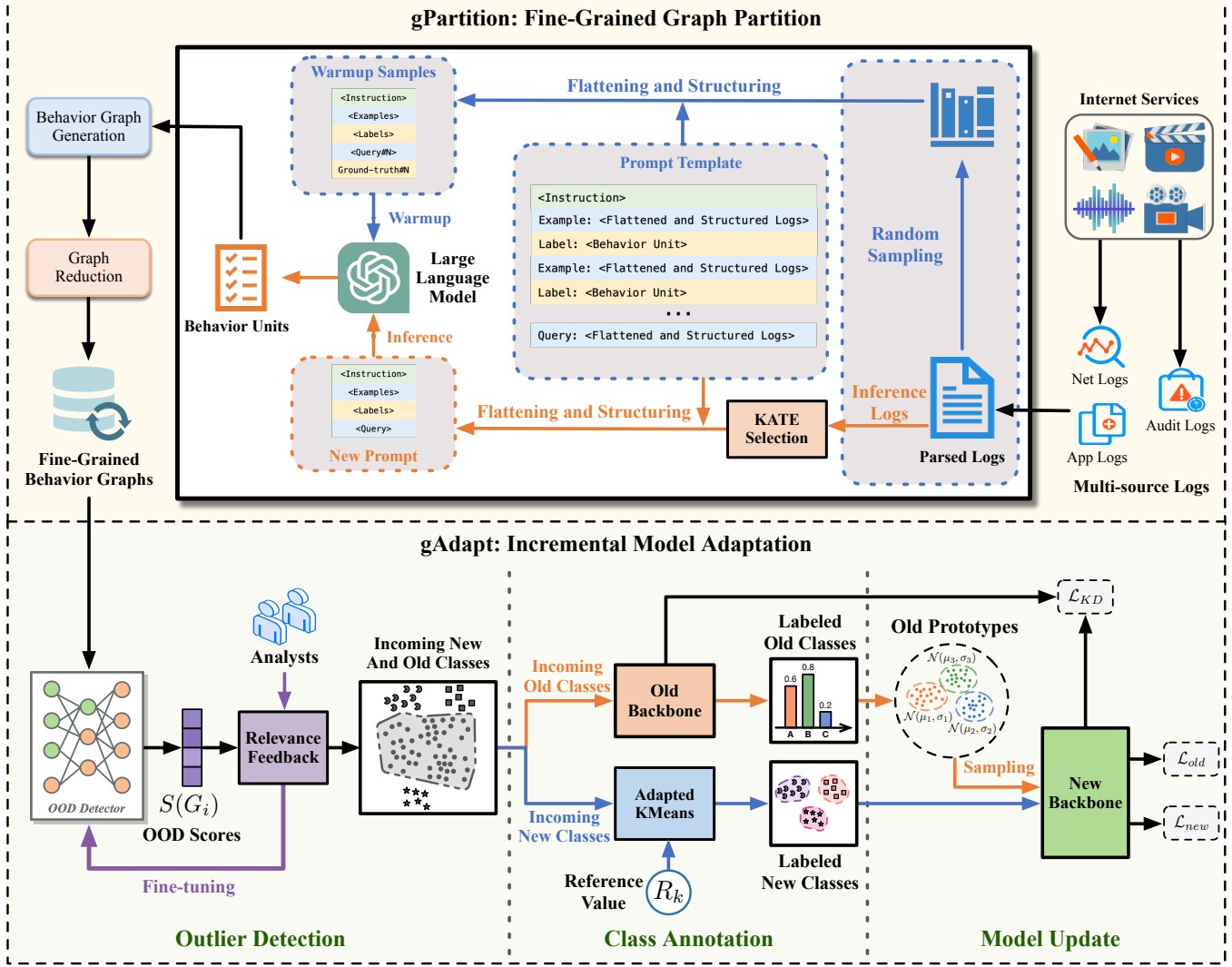

Figure 2: The main framework of the proposed FG-CIBGC.

no prior knowledge when suffering unprecedented behavior graphs in practice. When a novel class of behavior graphs emerges, we aim to proactively detect it and seamlessly incorporate it into the knowledge base, thereby strengthening the model's classification performance. Crucially, the updated model should be able to accurately identify previously unseen classes of behavior graphs in the future. This is a formidable challenge that eludes the capabilities of existing approaches.

## 4  Proposed Method

In this section, we present the proposed **F**ine-**G**rained and **C**lass-**I**ncremental **B**ehavior **G**raph **C**lassification (**FG-CIBGC**).

### 4.1  Overview

As shown in Fig. 2, the proposed FGCIBGC consists of two novel modules: gPartition and gAdapt. **First**, gPartition is proposed to learn the correlation of multi-source logs in a prompt and finish the

log correlation process. gPartition builds on the in-context learning paradigm with the large language models (LLMs). The derived correlation results, referred to as *behavior units*, are then used to partition the audit logs into fine-grained behavior graphs, which are subsequently reduced to achieve greater compactness. **Second**, gAdapt is proposed to proactively detect both known and unknown classes, classify them accurately, and correspondingly update the model in an incremental fashion. This is especially achieved through the synergistic combination of Out-Of-Distribution (OOD) detection and class-incremental graph learning.

### 4.2  gPartition: Fine-Grained Graph Partition

**Challenges Analysis.** Audit logs alone contain only low-level information about system activities, lacking the necessary knowledge to partition them into fine-grained operations. To accurately capture operations, the research communities observe that introducing logs with high-level semantics offers a more promising solution [33, 45]. Specifically, application logs of services are designed to record each

**Figure 3: The prompt templates omit insignificant log details for simplicity. Each prompt contains several labeled examples and one query. The last example in the prompt is the most similar to the query, whereas the first example is the least similar.**

operation and its attributes, while network logs explicitly track the corresponding network sessions incurred by each operation.

A vanilla method would be to leverage existing log correlation approaches to correlate these diverse log sources, allowing audit logs belonging to the same operations to be used in generating fine-grained behavior graphs. However, prior log correlation techniques rely on static correlation rules and overlook the potential for different logs to describe the same operation. This renders the use of logs with high-level semantics to partition audit logs into fine-grained behavior graphs a *fundamentally* new task.

**Rationale Behind gPartition.** To address this challenge of multi-source log correlation, we resort to the In-Context Learning (ICL) paradigm of large language models (LLMs), as LLMs have demonstrated superiority in log processing tasks. ICL with LLMs not only enables mining of the latent semantic correlations across multi-source logs, but also bolsters the advantages of rapid deployment and easy interactivity without large tuning costs. Additionally, we propose a novel type-position-aware prompt template specifically designed for log correlation tasks, coupled with a warmup mechanism that enhances the ICL capabilities of LLMs.

**Model Backbone.** The performance of the large language model is a key factor in the success of ICL. Considering that log messages are semi-structured sentences that are mainly composed of natural language descriptions (i.e., log template) [53], we chose GPT-3.5 [5], an LLM that is pre-trained on an extremely large amount of semantic information from the open-source corpus, as the backbone for gPartition. Recent large language models, including GPT-3.5, have demonstrated in-context learning (ICL) capabilities, motivating its use as the backbone for gPartition. As gPartition utilizes the LLM in a black-box manner, the backbone model can be replaced as long as the relevant API is accessible.

**Prompt Strategy.** Prompt strategy is the most significant part of ICL. To preprocess the influx of multi-source logs before prompting, we first parse the logs into standardized forms. After that, we merge these parsed logs in chronological order and segment them into batches of 400 entries. This approach considers that logs representing the same operation are typically temporally close, and a service's related operations do not correlate with an excessive number of logs.

Before designing the prompt template, an essential question arises: *How should we model log correlation task to facilitate understanding by LLMs?* To enable LLMs to perform the *"which-correlates-with-which"* task while adhering to the *"one query, one inference"* principle, we explicitly represent log types and position information,

transforming the problem into a compact format. Specifically, we have observed that position closeness is a crucial factor in log correlation, and different types of logs contain various fields that serve distinct roles in the correlating process. To address this, we insert two special tokens, namely <type> and <line#>, at each newline to indicate the log type and line ID, respectively, and subsequently flatten the log sequences into a linear format. Additionally, the instruction necessitates that the model provide the <type>|<line#> pairs to form a *behavior unit*, where each *behavior unit* represents the complete set of multi-source logs generated by a single service operation. This type-position-aware prompt format ensures both efficiency and accuracy in inference, as illustrated in Fig. 3.

It is important to note that the selection of examples in the prompt significantly impacts the downstream task performance of LLMs under the ICL paradigm. In this study, we utilize KATE [24] for in-context example augmentated selection. Due to page limit, the details of the selection algorithm are provided in the Appendix.

**Warmup Strategy.** Given that the model's ICL abilities can be enhanced through a warmup phase prior to ICL inference, we employ the following warmup process: gPartition first randomly selects 800 samples from the validation set to serve as prompt queries for the warmup. For each query, gPartition employs the aforementioned selection algorithm to identify the eight most similar samples in the training set as prompt examples, appending their ground-truth labels. These are then combined with fixed instructions to form a complete prompt. Subsequently, all prompts will be submitted to GPT-3.5 for parameter tuning in batches.

**Behavior Graph Generation.** Prior works showed the conversion of unstructured logs into a unified graph. Having obtained the behavior units, we implement an audit log parser to transform the audit logs within the same behavior units into a behavior graph, where each edge represents an audit log. This allows the audit logs to be partitioned into multiple small-scale behavior graphs. *However, not all audit logs belonging to the same operation can be incorporated into a behavior unit simply through log correlation.* For instance, system configuration activities of an operation may leave no traces in high-level semantics. To tackle this issue, we design a novel heuristic search algorithm to capture operations comprehensively. The overall workflow is shown in Alg 2.

**Graph Reduction.** Background noisy events resulting from the inherent low-level nature of auditing mechanism are massive, and they do not impact BGC task according to prior works [10, 17]. To reduce noisy events, we employ a few existing graph reduction algorithms including LogGC [20], CPR [43] and NodeMerge [38].

## 4.3 gAdapt: Incremental Model Adaptation

**Challenges Analysis.** A vanilla design would be to directly apply an existing class-incremental graph learning method to implement gAdapt. However, when deploying existing proposals in real-world scenarios, the research community faces *class increments* challenge that renders a straightforward design impractical. This is because the unknown class labels are inherently unpredictable, as the set of system operations is continuously expanding. Yet, the majority of existing methods tend to overlook this *open-world* problem.

In order to detect unknown classes, Out-Of-Distribution (OOD) Detection is showing promising potential recently. Therefore, a straightforward method is to use existing graph OOD detection method in an unsupervised way. However, they only finish *partial* of task requirements, that is, they only tell if a given behavior graph is OOD, but can not assign label to them so making it incapable of attaching new classes to the model's knowledge base. Besides, they do not consider the challenge of determining a threshold to decide if a datapoint is OOD.

**Rationale Behind gAdapt.** To address the aforementioned challenges, we implement gAdapt based on the idea that bridging the gap between OOD detection and class-incremental learning. We adopt a disentangled graph encoder as the model backbone, as graph formation typically follows a complex relational process, and such disentangled representation learning has shown promising results. Additionally, we employ replay-based methods for model updates. Upon receiving new samples, an OOD Detector generates OOD scores to identify *class increments*. The OOD samples are then clustered to obtain new-class labels, and used to update the model, alongside the structure incremental samples.

**Features Extraction.** We propose a novel strategy to extract node features for behavior graphs. In our analysis, we identified three distinct types of nodes within each behavior graph: processes, files, and sockets. Given the heterogeneous nature of these nodes, their respective feature vectors comprise different elements. For process nodes, we utilize [process_name, p_id, exe_name] as the feature set. File nodes are characterized by [file_name, inode, file_type], while socket nodes are represented by [ip, port, socket_type]. To encode all textual components within these feature sets, we employ FastText [4], a library for efficient learning of text representations.

**Outlier Detection.** We choose GOOD-D [25] as basic OOD Detector for its ability to detecting OOD graphs without using ground-truth labels. It is worth noting that OOD samples are often noisy. In real-world deployment, manual labeling is often required to enable custom configuration and adapt the model to new OOD samples [11]. However, this incurs nontrivial annotation cost.

With Weakly-supervised Relevance Feedback, we propose a method to overcome the challenge of custom configuration and human labeling. We introduce the hyperparameter $q$, which is a domain-interpretable value of the expected ratio of new OOD behavior graphs in the next time step. Specifically, we first apply the OOD detector to all inputs in the dataset $\mathcal{D}_t$ and compute their OOD scores $S(G_1), \ldots, S(G_n)$. These scores are sorted in ascending order, resulting in a permutation $\pi, S(G_{\pi(1)}), \ldots, S(G_{\pi(n)})$. Subsequently, we allow analysts to assign pseudo labels to the datapoints by selecting a domain-specific value for the hyperparameter $q$,

e.g. 0.05. The analysts label the top $q$ percent of the datapoints as OOD and the lower $1 - q$ percent as ID, receiving a labeled dataset for OOD detection feedback, i.e. $G_{\pi(1)} \ldots G_{\pi(n)}$ is labeled with $0_{\pi(1)}, \ldots, 0_{\pi(\lfloor n(1-q)\rfloor)}, 1_{\pi(\lceil n(1-q)\rceil)}, \ldots, 1_{\pi(n)}$. With this labeled dataset, we fine-tune the OOD detector. The introduced hyperparameter $q$ represents the ratio of OOD scores in the domain, which is not only an interpretable number but can also be determined with the help of prior knowledge without extensive tuning.

**Class Annotation.** For new-class samples, we utilize a K-Means [36] based clustering algorithm for class annotation due to its efficiency. Given the uncertainty in the number of unknown classes, naive K-Means, which requires a predefined cluster count, is not applicable. This prompts us to consider whether reference values exist for determining $K$. Fortunately, we observe that the lower bound of $K$ is defined by the distinct types of application logs, as limitations in logging tools and attack complexity prevent the capture of all execution details, leading similar logs to potentially reflect different behaviors. Thus, the number of different application log type can serve as the reference value $R_k$. Our goal is to find the optimal $R_k$ close to the reference value, evaluating the effectiveness using the Silhouette Score. The process is demonstrated in Alg 1.

---

**Algorithm 1:** Kmeans with parameter selection

  **Input:** Behavior graphs $G_i$ and behavior units $U_i$.
  **Output:** Behavior Clusters $C_1, C_2, C_3 \ldots C_{n_c}$.

1   $T \leftarrow \emptyset, s \leftarrow \emptyset$;
2   **for** $l_{app,j} \in U_i$ **do**
3     $\big\lfloor$   $T_{l_{app,j}} \leftarrow$ Log templates of application logs in $U_i$;
4   $n_{ref} \leftarrow$ Get_Reference_Value($T$);
5   **for** $i= n_{ref}$ *to* $3n_{ref}$ **do**
6     $\big\lfloor$   $s_{n_{ref}} \leftarrow$ Get_Silhouette_Score($G_i, n_{ref}$)
7   $n_c \leftarrow$ Choose_Optimal($s$)
8   $C_1, C_2, C_3 \ldots C_{n_c} \leftarrow$ Minibatch_Kmeans($G_i, n_c$);
9   Return $C_1, C_2, C_3 \ldots C_{n_c}$.

---

**Model Update.** We choose DisenGCN [28] as the backbone graph encoder considering its effectiveness and efficiency. Besides, to enhance model adaptability to structural and class increments, we adopt a replay-based incremental learning strategy utilizing class prototypes. Specifically, after stage $t - 1$, the old model has learned the optimal parameters $\Theta_{old}$. By feeding $\mathcal{D}_{t-1}$ into $\mathcal{M}^{old}$, we obtain old class embeddings. To overcome catastrophic forgetting, we construct class prototypes $\mathcal{N}(\mu_i, \sigma_i)$ to approximate $\mathcal{D}_{t-1}$, where $\sigma_i$ is the diagonal covariance. This reduces memory cost while preserving key information, as disentangled embeddings have most variance along the diagonal. For robustness, we use only correctly predicted samples to estimate $\mu_i, \sigma_i$.

In addition, since we only save the prototypes of the old data rather than the raw data, the old saved prototypes may not be available when the backbone is training on new data, i.e., the saved prototypes cannot represent the current positions of the old data in the embedding space. Therefore, when training the backbone $\mathcal{M}$ at stage $t$, we need to limit the shift of the old prototypes in the embedding space to ensure their availability. Thus, we add a loss to distill the knowledge of the old backbone $\mathcal{M}_{\Theta_{old}}$:

**Table 1: Overview of dataset for FG-CIBGC evaluation. We implement 8 attack scenarios based on their detailed reports of real-world APT campaigns [44]. This dataset consists of 4,992 graphs which can be categorized into 32 classes.**

| Scenarios | Attack Cases | #Graph | #Class | Avg of Node unreduced | Avg of Edge unreduced | Avg of Node reduced | Avg of Edge reduced | #Audit | #App | #Net |
|---|---|---|---|---|---|---|---|---|---|---|
| Apache | Data Leakage | 502 | 3 | 10.01 | 46.02 | 5.10 | 16.99 | 11.7MB | 43KB | 216KB |
| IM-1 | Data Leakage | 1,040 | 4 | 88.12 | 401.69 | 28.12 | 102.54 | 1.25GB | 52.1MB | 6.93GB |
| Vim | Unsafe Action | 125 | 3 | 28.6 | 1,256.00 | 14.6 | 52.32 | 432MB | 275KB | - |
| Redis | Unsafe Action | 201 | 2 | 14.00 | 451.19 | 8.23 | 66.54 | 63.78MB | 17KB | 987KB |
| Pgsql | Unsafe Action | 512 | 9 | 30.27 | 145.20 | 17.75 | 54.30 | 48.5MB | 360KB | 65.2MB |
| ProFTPd | Unsafe Action | 1,001 | 3 | 11.13 | 179.90 | 8.34 | 29.01 | 112.2MB | 95KB | 2.5MB |
| IM-2 | Unsafe Action | 1,040 | 4 | 65.68 | 1,360.25 | 35.68 | 114.96 | 796.8MB | 8.9MB | 3.94GB |
| Nginx | Misconfiguration | 1,001 | 4 | 5.14 | 17.93 | 3.00 | 10.51 | 9.8MB | 105KB | 133KB |

$$\mathcal{L}_{kd} := \mathbb{E}_{(\mathcal{G}, \mathcal{Y}) \sim \mathcal{D}_t} \left[ \| \mathcal{M}_{\Theta_{old}}(\mathcal{G}) - \mathcal{M}_{\Theta}(\mathcal{G}) \| \right]. \quad (1)$$

Besides, we use Prototype Augmentation (PA) [34] strategy to enhance the incremental learning backbone. Let $P_t$ denote the class prototypes before stage $t$ and $f_{PA}$ be the classifier after adding the classification heads of virtual classes. The loss function over old data is calculated by:

$$\mathcal{L}_{old} := \mathbb{E}_{(\mathcal{P}, \mathcal{Y}) \sim P_t} \left[ \mathcal{L}(f_{PA}(\mathcal{P}), \mathcal{Y})) \right]. \quad (2)$$

In addition, we use the following equation to calculate the classification loss on the new data:

$$\mathcal{L}_{cls,PA} := \mathbb{E}_{(\mathcal{G}, \mathcal{Y}) \sim D_t} \left[ \mathcal{L}(f_{PA}([\mathcal{M}_{\Theta}(\mathcal{G})]_{PA}, \mathcal{Y})) \right], \quad (3)$$

where $[\mathcal{M}_{\Theta}(\mathcal{G})]_{PA}$ represents the embeddings obtained by using $\mathcal{M}_{\Theta}(\mathcal{G})$ after the Prototype Augmentation step. Finally, we have the total loss function as follows:

$$\mathcal{L} = \mathcal{L}_{cls,PA} + \alpha * \mathcal{L}_{old} + \beta * \mathcal{L}_{kd}, \quad (4)$$

where $\alpha, \beta$ are used to balance $\mathcal{L}_{cls,PA}, \mathcal{L}_{old}$ and $\mathcal{L}_{kd}$.

## 5 EXPERIMENT SETUP

In this section, we introduce the experimental setup, including the datasets, baselines, evaluation metrics and implementation details.

### 5.1 Datasets

In order to support the thorough evaluation of FG-CIBGC, the dataset should have the following properties:

- **Completeness of Log Sources.** The dataset should offer complete log sources including application logs and network logs. Without them, behavior graph identification relies either on a static knowledge base or search within a pre-obtained graph, both of which are impossible in a class-incremental setting.
- **Diversity of Behavior Types.** The dataset should contain a diverse range of system behaviors. Limited types of behavior graphs restrict thorough evaluation.

Open-source datasets like DARPA Trace [6] and StreamSpot [29] lack application and network logs *completely*. Moreover, they also fail to cover diverse operations in Internet Infrastructure. Given the above limitations, we construct a new behavior dataset that satisfies both properties, featuring 3 log types and 4,406 graphs across 31 classes. The statistics of the dataset is shown in Table 1. Due to page limit, more details are presented in the Appendix.

### 5.2 Baselines

To facilitate a comprehensive evaluation, we compare the proposed FG-CIBGC framework with existing methods from two key perspectives: **(i) Behavior Graph Classification Performance. (ii) Efficacy in Attack Investigation**.

**(i) Behavior Graph Classification Performance.** In terms of behavior graph classification task, we use 12 existing methods as baselines, covering state-of-the-art methods in the behavior graph classification landscape. (i) Three behavior graph classification methods: **Tgminer** [54], **Watson** [46] and **DepComm** [42].(ii) Six class-incremental incremental learning methods: **EWC** [19],**LwF** [21], **GEM** [27], **TWP** [22], **CPCA** [34] and **Fine-Tuning** [3]. (iii) Three graph-level dynamic graph learning methods: **tdGraphEmbed** [2], **GraphERT** [1] and **TP-GNN** [23].

**(ii) Efficacy in Attack Investigation.** The fine-grained behavior graph classification task is inherently designed to facilitate downstream applications. Thus, we analyze whether the generated behavior graphs can benefit downstream tasks. Specifically, we select attack investigation as the representative downstream task, given its practical significance. In brief, the attack investigation task aims to identify attack-related edges within a given behavior graph. We use **Watson** [46] and **DepComm** [42] which generate coarse-grained behavior graphs as baselines. Besides, we employ **DepImpact** [8] as the baseline for attack investigation.

### 5.3 Evaluation Metrics

Metrics such as F1-Score (F1) have been adopted in prior studies to evaluate the behavior graph classification task. Following the convention, we use F1 as evaluation metric to conduct the experiments on behavior graph classification. However, none of these metrics take into account the fine-grained requirements of behavior graph classification. Considering such fine-grained characteristics could enable a fair comparison between coarse-grained and fine-grained behavior graph classification methods. In this regard, borrowing the idea from the MIoU (Mean Intersection over Union) metric in semantic segmentation, we propose a new metric called EIoU (Edge Intersection over Union). The EIoU metric enables a comprehensive evaluation of different methods by capturing fine-grained requirements. Specifically, EIoU reframes the graph classification problem as an edge-level classification task, where the classification of a graph into a specific category corresponds to the assignment of its edges to that category. By applying matching criteria to the

Table 2: Comparison results (EIoU % and F1 %) of fine-grained class-incremental behavior graph classification task across all datasets. "(+)" indicates that the input to this baseline is the fine-grained behavior graphs generated by gPartition. The best results are shown in bold type and the runner-ups are underlined.

| Method | Apache | | IM-1 | | IM-2 | | Vim | | Redis | | Pgsql | | ProFTPd | | Nginx | |
|---|---|---|---|---|---|---|---|---|---|---|---|---|---|---|---|---|
| | EIoU | F1 | EIoU | F1 | EIoU | F1 | EIoU | F1 | EIoU | F1 | EIoU | F1 | EIoU | F1 | EIoU | F1 |
| Tgminer | 48.82 | 65.15 | 59.64 | 66.33 | 52.63 | 67.67 | 58.72 | 60.13 | 64.21 | 79.06 | 51.72 | 66.43 | 59.44 | 59.75 | 53.23 | 65.91 |
| Watson | 40.24 | 69.23 | 42.62 | 56.34 | 39.67 | 66.30 | 47.09 | 56.79 | 62.35 | 82.65 | 48.27 | 61.16 | 41.22 | 52.25 | 44.74 | 54.86 |
| DepComm | 41.09 | 70.54 | 51.39 | 75.99 | 43.61 | 72.51 | 54.80 | 72.84 | 60.71 | 80.80 | 52.75 | 70.45 | 57.70 | 67.91 | 57.56 | 76.46 |
| Fine-Tuning(+) | 50.84 | 71.08 | 52.63 | 76.35 | 55.93 | 74.94 | 56.05 | 73.75 | 64.23 | 78.97 | 56.24 | 73.27 | 54.38 | 68.21 | 57.41 | 78.92 |
| EWC(+) | 55.81 | 74.95 | 54.60 | 78.02 | 61.90 | 80.74 | 64.10 | 82.42 | 69.20 | 85.15 | 68.49 | 87.32 | 67.43 | 84.34 | 63.62 | 80.84 |
| LwF(+) | 62.32 | 83.22 | 58.46 | 77.83 | 59.75 | 76.85 | 62.80 | 76.10 | 64.21 | 85.84 | 57.29 | 81.61 | 66.46 | 87.26 | 63.54 | 82.23 |
| GEM(+) | 58.74 | 76.83 | 58.98 | 74.98 | 59.86 | 76.93 | 68.20 | 82.15 | 64.53 | 84.43 | 58.67 | 81.26 | 57.72 | 81.03 | 64.27 | 83.53 |
| TWP(+) | 69.42 | 86.88 | 65.64 | 85.55 | 67.04 | 86.57 | 72.08 | 84.43 | 70.12 | 85.43 | 50.74 | 76.95 | 70.16 | 85.46 | 61.64 | 85.89 |
| CPCA(+) | 67.08 | 86.42 | 59.94 | 81.91 | 67.63 | 87.84 | 71.27 | 83.26 | 70.29 | 87.24 | 67.93 | 85.54 | 71.09 | 87.82 | 71.60 | 87.60 |
| tdGraphEmbed(+) | 56.43 | 75.12 | 46.43 | 60.78 | 48.25 | 67.37 | 56.26 | 73.81 | 59.52 | 79.03 | 53.68 | 72.62 | 51.48 | 67.63 | 46.48 | 61.54 |
| GraphERT(+) | 67.62 | 77.71 | 52.36 | 68.44 | 58.69 | 78.23 | 65.71 | 75.32 | 64.52 | 78.39 | 64.15 | 79.42 | 62.08 | 77.89 | 55.37 | 75.32 |
| TP-GNN(+) | 62.32 | 78.56 | 55.32 | 71.84 | 57.36 | 77.93 | 64.08 | 74.26 | 63.12 | 76.21 | 59.34 | 73.45 | 53.48 | 72.09 | 60.02 | 79.84 |
| **Ours** | **74.62** | **91.62** | **70.35** | **90.26** | **73.93** | **93.92** | **76.08** | **96.07** | **78.24** | **98.32** | **74.56** | **93.08** | **74.28** | **92.73** | **73.31** | **93.27** |

ground truth at each fine-grained category, we construct a confusion matrix delineating True Positives (TP), True Negatives (TN), False Positives (FP), and False Negatives (FN). With these definitions in place, Intersection over Union (IoU) can be formulated by $\text{IoU} = \frac{|TP_e|}{|FP_e| + |TP_e| + |FN_e|}$, where $|TP_e|$, $|FP_e|$, and $|FP_e|$ respectively stand for the number of edge-level TP, FP, and FN. For the EIoU metric used in behavior graph classification, the matching criterion is determined by the accurate edge-level prediction corresponding to the ground truth fine-grained labels. The EIoU is calculated as:

$$\text{EIoU} = \sum_{i=0}^{k} \text{IoU}_i, \tag{5}$$

where $\text{IoU}_i$ represents the IoU of fine-grained class $i$ and $k + 1$ is the total number of fine-grained classes in the evaluated dataset.

To quantify the attack investigation performance, we treat the task as an edge-level binary classification problem, as attack investigation inherently aims to identify attack-related edges. Consequently, we compute the Accuracy (Acc) and F1-Score (F1) to evaluate the attack investigation tasks.

## 5.4 Implementation Details

We prototype FG-CIBGC in 42K lines of Python code. The proposed model is implemented by PyTorch 2.1 framework on Ubuntu 22.04, and all the evaluations are conducted on NVIDIA GeForce RTX 3090 card. For a fair comparison, we tune the hyper-parameters of the base Class-Incremental learning model using grid-search: learning rate $lr \in \{0.005, 0.001, 0.01\}$, batch size $b \in \{512, 1024, 2048\}$, embedding dimension $d \in \{32, 64, 128, 256\}$. We set $C_k : C_u$ (representing the number of known and unknown classes) $= 9 : 1$.

## 6 RESULTS AND ANALYSIS

In this section, we conduct experiments regarding behavior graph classification performance, ablation study, efficacy in attack investigation and hyper-parameter sensitivity to validate the proposed FG-CIBGC. Due to the page limit, we have to move additional results, including but not limited to more ablation study results and the associated time analysis to the Appendix.

Table 3: Ablation study results. The best results are shown in bold type and the runner-ups are underlined.

| Method | Apache | | IM-1 | | IM-2 | |
|---|---|---|---|---|---|---|
| | EIoU | F1 | EIoU | F1 | EIoU | F1 |
| Baseline | 43.95 | 71.93 | 54.36 | 73.21 | 48.56 | 77.21 |
| Baseline-T | 50.95 | 77.93 | 55.36 | 76.21 | 65.56 | 82.21 |
| Baseline-C | 67.08 | 86.42 | 59.94 | 81.91 | 67.63 | 87.84 |
| **FG-CIBGC** | **74.62** | **91.62** | **70.35** | **90.26** | **73.93** | **93.92** |

### 6.1 Behavior Graph Classification Performance

In this section, we compare the behavior graph classification performance of FG-CIBGC with the constructed baselines. The results are shown in Table 2. By observing the experimental results, we can have the following observations:

(1) Coarse-grained behavior graph-based baselines generally exhibit relatively low EIoU performance. WatSon performs an adapted DFS on every single data object found in the KG, except for libraries that do not reflect the roots of user-intended goals. DepComm identifies process-centric communities. They all fail to find operations centered around data/process objects, leading to undesirable performance. Among all coarse-grained behavior graph-based methods, Tgminer emerges as the top performer in terms of the EIoU metric. This can be attributed to its strategic focus on finding frequent patterns, rather than centering around data or processes, which sets it apart from other methods. In contrast, FG-CIBGC excels by leveraging more comprehensive information from diverse sources, resulting in accuracy in identifying behavior boundaries.

(2) FG-CIBGC demonstrates significant superiority over class-incremental graph learning baselines. These baselines struggle to accommodate scenarios with unknown new classes, and thus fail to adapt effectively. Furthermore, graph-level dynamic graph learning baselines lag behind class-incremental learning methods, as they lack the ability to adapt to known new classes.

(3) FG-CIBGC outperforms baselines across all datasets, achieving an average improvement of 4.89% in EIoU and 6.82% in F1-Score

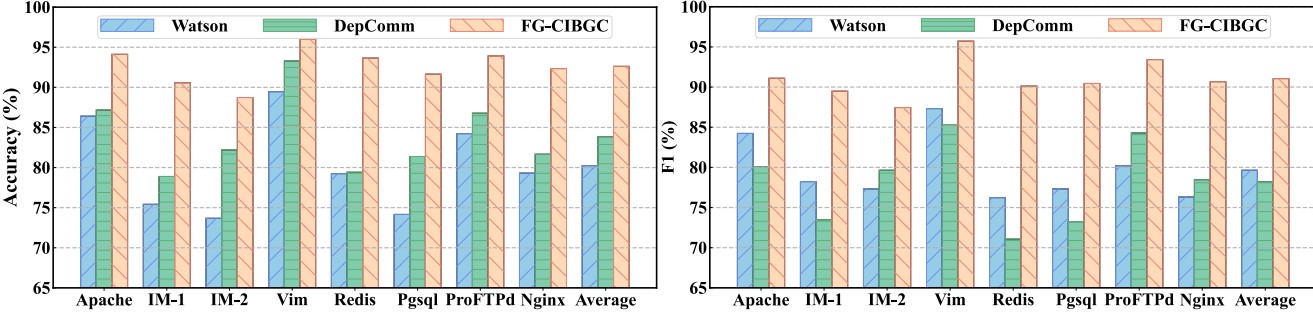

**Figure 4: Experimental results regarding effect on attack investigation task.**

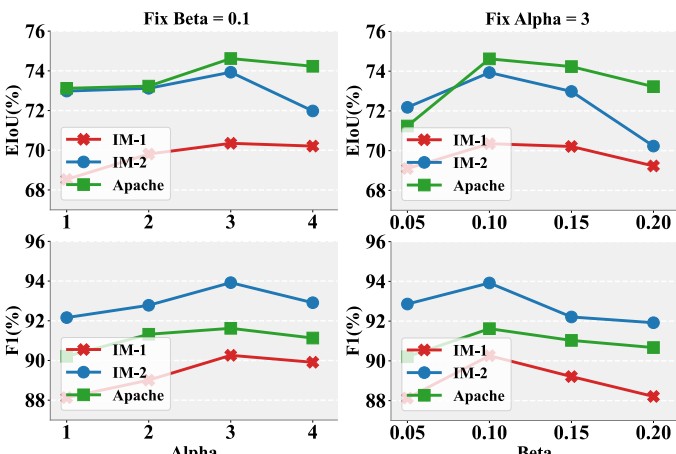

**Figure 5: Hyper-parameter sensitivity analysis results for different values of hyper-parameters $\alpha$ and $\beta$ on Apache, IM-1 and IM-2 datasets.**

compared to the baselines. The superiority is largely attributed to the combination of the innovative gPartition and gAdapt components. Therefore, FG-CIBGC excels in fine-grained and class-incremental behavior graph classification task.

## 6.2 Ablation Study

The proposed FG-CIBGC framework contains two major components. We conduct an ablation study on 3 representative datasets to further verify their effectiveness. Specifically, 4 combinations of key modules are compared in the ablation study as follows:

- **Baseline:** For this variant, we employ Tgminer to identify behavior graphs, and leverage CPCA to perform incremental graph classification.
- **Baseline-T:** For this variant, we substitute the gPartition component of FG-CIBGC with the behavior graph generation algorithm of Tgminer.
- **Baseline-C:** For this variant, we replace the gAdapt component of FG-CIBGC with the class-incremental learning baseline CPCA.
- **FG-CIBGC:** This variant is the proposed FG-CIBGC model.

As shown in Table 3, we can draw the following conclusions:

(1) The 'Baseline' performs the worst due to its inability to generate behavior graphs matching service operations. Furthermore, it lacks an efficient strategy to detect known classes.

(2) Incorporating gPartition or gAdapt into the 'Baseline' markedly enhances its performance, highlighting the necessity of generating fine-grained behavior graphs and combining OOD detection with class-incremental learning.

(3) The two proposed modules achieve stable and effective performance on different datasets. FG-CIBGC leverages the advantages of its modules to achieve significant performance gains.

## 6.3 Efficacy In Attack Investigation

In this section, we seek to ascertain whether the proposed FG-CIBGC can indeed generate fine-grained behavior graph classification results that are effective in facilitating the downstream attack investigation task. We utilize **DepImpact** to identify critical components in a unified graph derived from raw audit logs. FG-CIBGC and baselines partition the unified graph and classify the resulting behavior subgraphs, which guide forward and backward causality analysis. As illustrated in Fig. 4, it is evident that FG-CIBGC demonstrates the best performance in fine-grained behavior graph generation, thereby optimally facilitating attack investigation.

## 6.4 Hyper-Parameter Sensitivity

In this section, we perform hyper-parameter sensitivity analysis on 3 representative datasets to investigate the impact of $\alpha$ and $\beta$ on FG-CIBGC by conducting a grid search for their optimal values. Initially, we set $\beta = 0.3$ and vary $\alpha$, followed by fixing $\alpha = 1$ while varying $\beta$. The experimental results are illustrated in the Fig. 5. Overall, FG-CIBGC maintains solid performance with different parameter settings. The optimal performance is observed when $\alpha = 3$ and $\beta = 0.1$, indicating its strongest capability under this setting.

## 7 CONCLUSION

This paper presents FG-CIBGC, the first unified framework for fine-grained and class- incremental behavior graph classification. FG-CIBGC comprises two novel modules: gPartition for fine-grained graph partitioning, and gAdapt for unknown class detection and adaptation. To validate its efficacy, we introduce a novel benchmark. This benchmark includes a new dataset of 4,992 graphs across 32 classes, derived from 8 attack scenarios. It also features a novel Edge Intersection over Union (EIoU) evaluation metric. Extensive experiments demonstrate FG-CIBGC's superior performance on fine-grained and class-incremental behavior graph classification tasks. Furthermore, FG-CIBGC has the ability to generate fine-grained behavior graphs that facilitate downstream applications.

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

## A  Overview

We provide in this Appendix that cannot fit into the main paper due to the page limit, including **More Ablation Studies** on the variants of FGC-CIBGC, the details of the **Heuristic Search Algorithm**, **More Hyper-Parameter Analysis** regarding learning rate, batch size, and hidden size, **Complexity Analysis** of the model. We also give detailed descriptions on **Datasets**, **Code File**, and **Further Explanation of Implementation**.

## B  More Ablation Studies

In this section, we seek to substantiate the rationale behind our choice of the GPT-3.5 model. For the purpose of comparison, we selected Codex, GPT-3 and LLaMa-2 as baselines. Results are summarized in Table. 4.

It can be seen that FG-CIBGC has a relatively large accuracy advantage when using Codex as the backbone compared to other LLMs. The results indicate that the use of an appropriate baseline model as the backbone greatly influences FG-CIBGC's performance.

## C  The Search Algorithm

In this section, we present the details of the heuristic search algorithm. For the unified graph without partitioning, we first retrieve the time ranges of the edges belonging to the same behavior unit

---

**Algorithm 2:** Workflow of heuristic search algorithm

**Input:** The Unified graph $\mathcal{G}_u$, in which some audit events are correlated with behavior units.

**Output:** Behavior graphs $\mathcal{G}$.

1 $T \leftarrow$ Time spans of all audit events (edges);
2 $\mathcal{G} \leftarrow$ Get_Components($\mathcal{G}_u, T$);
3 **for** $G_i \in \mathcal{G}$ **do**
4    $\mathcal{E}_i \leftarrow$ Entities included in $G_i$;
5    **for** *node* $v \in \mathcal{E}_t^i$ **do**
6       $E_{\text{out}} \leftarrow$ Uncorrelated outgoing edges of node $v$ in $\mathcal{G}_u$;
7       $E_{\text{in}} \leftarrow$ Uncorrelated incoming edges of node $v$ in $\mathcal{G}_u$;
8       **if** $E_{out} \neq \emptyset$ *or* $E_{in} \neq \emptyset$ **then**
9          $E_{\text{unalloc}} \leftarrow E_{\text{out}} \cup E_{\text{in}}$;
10          **for** *edge* $e \in E_{unalloc}$ **do**
11             $\mathcal{G}_{\text{closest}} \leftarrow$ Get_Closest($e, \mathcal{G}$)
12             $\mathcal{G}_{\text{closest}} \leftarrow \mathcal{G}_{\text{closest}} \cup \{e\}$;

13 Return $\mathcal{G}$.

---

in the unified graph. We then compute the minimum connected components on the unified graph that contain these audit log edges within the respective time ranges. Finally, we inspect each node in the connected components to identify any edges that are not assigned to higher-level behavior units. We allocate such edges to the most recent behavior unit. The overall process is shown in Alg. 2.

## D  More Comparison Experiments

Building upon our previous work, we have further incorporated Accuracy (Acc) as an evaluation metric, resulting in the tabular format shown in Table. 7. Based on the experimental results, we can observe that FG-CIBGC significantly outperforms the comparative methods in terms of the accuracy metric.

## E  Datasets

In this section, we will briefly introduce the eight datasets we use. First, we will provide a brief overview of the CVE vulnerabilities. In addition, we will present methods for data collection, data cleaning and preprocessing, along with corresponding labeling rules for each dataset.

### E.1  Basic Information

In this section, we will briefly introduce the basic information of each dataset, as well as the CVE vulnerabilities. In this work, we not only consider web applications, but also explore a broader range of scenarios including file editing, file transfer, and Internet applications. Our goal is to validate the effectiveness of our method across a diverse set of real-world use cases, beyond just web-based applications.

By evaluating our approach in this expanded application scope, we aim to demonstrate its generalizability and robustness. The

**Table 4: Ablation study results (EIoU % and F1 %) of different LLM Backbones. The best results are shown in bold type and the runner-ups are underlined.**

| Method | Apache | | IM-1 | | IM-2 | | Vim | | Redis | | Pgsql | | ProFTPd | | Nginx | |
|---|---|---|---|---|---|---|---|---|---|---|---|---|---|---|---|---|
| | EIoU | F1 | EIoU | F1 | EIoU | F1 | EIoU | F1 | EIoU | F1 | EIoU | F1 | EIoU | F1 | EIoU | F1 |
| Codex | 68.52 | 82.15 | 58.25 | 84.33 | 50.03 | 84.67 | 56.01 | 59.13 | 63.21 | 89.06 | 60.34 | 86.43 | 59.44 | 59.75 | 70.23 | 85.91 |
| GPT-3 | 60.24 | 68.23 | 59.23 | 86.34 | 49.78 | 89.32 | 67.39 | 86.25 | 69.35 | 89.65 | 68.27 | 81.16 | 70.22 | 88.65 | 64.74 | 88.39 |
| LLaMa-2 | 51.09 | 85.54 | 51.39 | 80.25 | 48.61 | 76.51 | 60.80 | 84.85 | 66.21 | 86.80 | 62.75 | 71.95 | 61.70 | 67.91 | 57.56 | 80.46 |
| **Ours** | **74.62** | **91.62** | **70.35** | **90.26** | **73.93** | **93.92** | **76.08** | **96.07** | **78.24** | **98.32** | **74.56** | **93.08** | **74.28** | **92.73** | **73.31** | **93.27** |

inclusion of file management, communication, and content creation tasks allows us to assess the performance of our technique in contexts that go beyond typical web applications.

This comprehensive evaluation strategy enables us to thoroughly verify that our method can achieve strong results across a wide spectrum of application domains, rather than being limited to a narrow set of web-centric scenarios. The breadth of the tested scenarios strengthens the practical significance and impact of our contributions.

We have selected applications that are critical for building the Internet services.

(1) Apache Vulnerability : CVE-2021-41773

Apache is an open-source, cross-platform web server software and stands as one of the most widely-used choices for Internet services on the web today. It is developed and maintained by the Apache Software Foundation. Apache, renowned for stability, security, and efficiency, supports various operating systems including Windows, Linux, etc. It provides rich features and flexible configuration options, making it suitable for building various types of websites, including static websites, dynamic websites, and web applications. Apache supports a variety of programming languages and technologies, including PHP, Python, Perl, CGI, and more, empowering developers to effortlessly craft robust web applications. Moreover, Apache supports advanced features like virtual hosting, SSL/TLS encryption, URL rewriting, etc., catering to diverse requirements in website construction and operation.

Apache HTTPd Server 2.4.49 version introduces a new function with a path traversal vulnerability, but it needs to be combined with the traversal directory configuration "Require all granted". Attackers can exploit this vulnerability to achieve path traversal, read arbitrary files, or execute bash commands in httpd programs configured with CGI, thereby gaining the opportunity to control the server and access the root directory to read the files inside.

(2) Apache-Pgsql Vulnerability : CVE-2019-9193

PostgreSQL is one of the most popular database systems today. It is the most commonly used database on Mac OSX systems, but it also provides versions for Windows and Linux operating systems. (Metasploit on Kali uses the PostgreSQL database.)

The vulnerability is caused by a feature of PostgreSQL that allows specific users to execute arbitrary code within the PostgreSQL environment. This feature is enabled by default in PostgreSQL versions 9.3-11.2. Starting from version

9.3, PostgreSQL implemented a new feature called COPY TO/FROM PROGRAM, which allows superusers and users to execute arbitrary operating system commands.

In this dataset, we set up a web service using Apache and PostgreSQL version 9.3. We utilized the vulnerability to perform normal database operations and attack operations. We collected application logs, audit logs, and network logs from Apache and PostgreSQL.

(3) IM-1 Vulnerability : CVE-2016–3714

ImageMagick is a widely used image processing program that many vendors use for tasks such as resizing, cropping, watermarking, and format conversion. However, researchers have discovered that when a user inputs an image containing 'malformed content,' it can trigger a command injection vulnerability. One of the most serious vulnerabilities is CVE-2016-3714, which allows remote code execution. This vulnerability affects version 6.9.3-9 and all versions prior.

ImageMagick has a feature called delegate, which is used to call external libraries to handle files. The process of calling external libraries uses the system's system command, which is the cause of this vulnerability.

(4) IM-2 Vulnerability : CVE-2022-44268

In ImageMagick versions prior to 7.1.0-51, there is a feature in the code that handles PNG files. This feature can lead to the reading of arbitrary files on the current operating system when converting images, and then outputting the contents of those files into the image content.

In the above two datasets of ImageMagick, we constructed a web application using Apache and ImageMagick, and collected logs from multiple sources.

(5) Nginx Vulnerability : Path Traversal

Nginx is a high-performance open-source web server and reverse proxy server known for its exceptional performance and high reliability. Nginx uses an event-driven architecture and asynchronous non-blocking processing to handle a large number of concurrent connections, making it perform well under high loads. Nginx also offers flexible configuration options and rich features, making it suitable for various types of web services, including serving static content, dynamic content, and reverse proxying. Due to its low resource consumption, high stability, ease of configuration, and scalability, Nginx has become the preferred server software for many websites and applications.

In Nginx, when configuring an alias using the alias directive, forgetting to include a trailing slash (/) (i.e., using "/files"

instead of "/files/") can result in a directory traversal vulnerability. The original purpose of this location block was to allow users to access files under the /home/ directory.

(6) ProFTPD Vulnerability : CVE-2019-12815

ProFTPD is an open-source, highly configurable FTP server software that supports multiple operating systems, including Linux, Unix, and Windows. It offers rich features and flexible configuration options to meet various FTP server requirements. ProFTPD is known for its good performance and security, supporting virtual users, restricting user permissions, logging, and SSL/TLS encrypted transmission for data security. Easy to install and configure, ProFTPD is suitable for networks of all sizes and is a popular choice for many organizations and individuals as an FTP server software.

There is a vulnerability in ProFTPD <= 1.3.6 that allows arbitrary file copying. This vulnerability is due to the custom SITE CPFR and SITE CPTO commands in the mod_copy module not properly checking read/write permissions. An attacker can exploit this vulnerability to copy any file on the FTP server without permission.

(7) Redis Vulnerability : CVE-2022-0543

Redis is an open-source in-memory database that can also be used as a cache and message broker. It supports various data structures such as strings, hashes, lists, sets, and sorted sets, providing rich commands and flexible configuration options. Redis offers high performance, persistence, replication, clustering, and more, making it versatile across various applications. As an efficient key-value store and caching solution, Redis is widely used in the Internet and big data fields. Its simplicity and ease of use allow developers to quickly build high-performance applications and provide reliable data storage and access services in distributed environments.

Redis has a vulnerability where, after a user connects to Redis, they can execute Lua scripts using the eval command. However, these scripts are run in a sandbox, and under normal circumstances, cannot execute commands or read files. Some distributions, such as Debian, Ubuntu, and CentOS, patch the original software with additional packages. For example, Debian's patch for Redis includes an include statement.

Unfortunately, in Debian and Ubuntu's packaging of Redis, a package object was inadvertently left in the Lua sandbox. Attackers can exploit this object to load functions from the liblua dynamic link library (DLL) and escape the sandbox to execute arbitrary commands. By utilizing the package.loadlib function in the Lua sandbox to load functions from /usr/lib/x86_64-linux-gnu/liblua5.1.so.0, an attacker can gain access to the io library and use it to execute commands.

(8) Vim Vulnerability : CVE-2019-12735

Vim is a powerful text editor widely used on Unix and Unix-like systems. It boasts advanced features such as syntax highlighting, code folding, auto-completion, multi-level undo/redo, and macro recording. Vim supports various modes of operation, including insert mode, command mode, and visual mode, making editing more efficient. Vim also supports a variety of plugins and scripts to extend its functionality. Due to its high degree of customization and powerful features, Vim is favored by many developers and system administrators as their preferred editor.

CVE-2019-12735 is a vulnerability in Vim versions before 8.1.1365 and Neovim versions before 0.3.6. It allows remote attackers to execute arbitrary OS commands via the :source! command in a modeline, as demonstrated by execute in Vim, and assert_fails or nvim_input in Neovim.

## E.2 Data Collection

For each dataset, the data collection process involves gathering application logs, network logs, and audit logs. We employ Linux Auditd, a widely used tool for recording audit logs [13], along with built-in logs from Internet services for application logs, while capturing network logs with tshark. It should be noted that log collection configurations must be adjusted to capture comprehensive fields, as default settings only gather a limited set, resulting in incompleteness. In addition, we outline the specific collection methods for these three types of logs [31, 33, 45].

(1) Application Logs

Different kinds of applications employ distinct logging mechanisms tailored to their specific needs. Application logs record important events with application-specific semantics pertaining to the application's behavior, errors, and performance. To collect application logs, we conducted research on the optional configurations of various application logs, aiming to record all log fields that are useful for the experiment to the fullest extent. We focused on fields such as IP addresses, port numbers, payloads, and fields that can reveal the type of event. This enabled us to achieve full collection of application logs.

We have configured the application logs for Apache and PostgreSQL with special configurations, while the rest of the application logs use default configurations.

Apache:

```
1  $LogFormat  "%h  %l  %u  %t  \"%r\"  %>s
2  %O  %a  %A  %p  %P"  combined
3  $CustomLog  /var/log/apache2/access.log
4  combined
```

Postgresql:

```
1  #  -  Where  to  Log  -
2  log_destination  =  'stderr,csvlog'
```

(2) Network Logs

Network logs are records of network traffic, detailing communication between devices. They contain valuable information such as source and destination IP addresses, ports, protocols, and timestamps.

Tshark, a command-line network protocol analyzer, is a powerful tool for capturing and analyzing network logs. It can capture live traffic from a network interface or read saved capture files. Tshark's filtering capabilities allow analysts to focus on specific traffic of interest, making it easier

to identify patterns or anomalies. Additionally, Tshark can output captured data in various formats for further analysis or integration with other tools. Overall, Tshark is a versatile tool for analyzing network logs and gaining insights into network activity. Therefore, We collect network logs containing all fields using tshark, saving and reading them in JSON format.

We have applied a uniform tshark configuration to all network logs as follows:

```
1  $tshark -n -r test.pcap -T fields -E
2  header=y -e frame.number
3  -e frame.time
4  -e ip.src -e tcp.srcport
5  -e ip.dst -e tcp.dstport
6  -e ip.proto -e frame.len
7  -e _ws.col.Info
8  -e frame.interface_name
9  -e frame.interface_description
10 -e frame.encap_type
11 -e frame.offset_shift
12 -e frame.time_epoch -e frame.time_delta
13 -e frame.time_delta_displayed
14 -e frame.time_relative
15 -e frame.cap_len -e eth.dst -e eth.src
16 -e eth.type -e http.response.version
17 -e http.response.code
18 -e http.response.code.desc
19 -e http.response.phrase -e http.server
20 -e http.response.line
21 -e http.content_encoding
22 -e http.content_length
23 -e http.connection -e http.content_type
24 -e http.response_number
25 -e http.time
26 -e http.request_in
27 -e http.response_for.uri
28 -e http.file_data
29 -e data-text-lines
30 > out.tsv
```

(3) Audit Logs

Audit logs are records that provide a detailed account of system and application activity, helping organizations track access, changes, and other events for security and compliance purposes. Auditd is the user-space component of the Linux Auditing System, responsible for writing audit records to the disk. It monitors various system calls and generates audit logs based on pre-defined rules.

Auditd allows administrators to configure what events to monitor and how to handle them. It can log events such as file access, process execution, user authentication, and more. The audit logs produced by Auditd are stored in a binary format and can be viewed using the ausearch or aureport commands.

auditd provides detailed information about system activity, helping administrators detect unauthorized access attempts, track system changes, and investigate security incidents. It is a critical component of a comprehensive security monitoring strategy for Linux systems, offering insights into system behavior and helping ensure compliance with security policies and regulations.

We use Auditd to record the system calls related to processes and files involved in the experiment. Then, we manually analyze the data to remove redundant system call information.

For audit logs, we have applied special configurations to Pgsql, IM-1, and IM-2 datasets, while the rest of the datasets use default configurations.

Apache-Pgsql:

```
1  -D
2  -b 8192
3  -f 1
4  --backlog_wait_time 0
5  -a always,exit -S all -F exe=/usr/local/
6  pgsql/bin/postgres -k pgsql_audit
7  -w /etc/passwd -p rwxa -k passwd_audit
8  -a always,exit -S all -F exe=/usr/sbin/
9  apache2 -k apache_audit
10 -a always,exit -F arch=b64 -S bind
11 -a always,exit -S read,write,open,close,
12 clone,fork,vfork,execve,kill,
13 mq_open,openat,sendto,recvfrom,sendfile,
14 sendmsg,sendmmsg,recvmsg,recvmmsg,
15 connect,socket,unlink,link,linkat,
16 unlinkat,rmdir,mkdir,reename,
17 pipe,pipe2,dup,dup2,getpeername,
18 fcntl
```

IM-1:

```
1  -D
2  -b 8192
3  --backlog_wait_time 0
4  -f 1
5  -a always,exit -S fstat,getsockname,
6  connect,read,close,
7  shutdown,sendto,
8  recvfrom,openat,writev,write,bind,
9  unlink -F exe=/usr/sbin/apache2
10 -k apache_audit
11 -a always,exit -S openat,execve,read,
12 write,close -F exe=/usr/bin/id
13 -k monitor_id
14 -w /etc/passwd -p rwxa -k passwd_audit
15 -a always,exit -S clone -F exe=/bin/sh
16 -k monitor_sh
17 -a always,exit -S all -F exe=/usr/local/
```

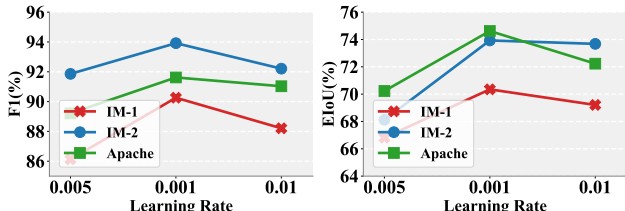

Figure 6: Ablation study on learning rates.

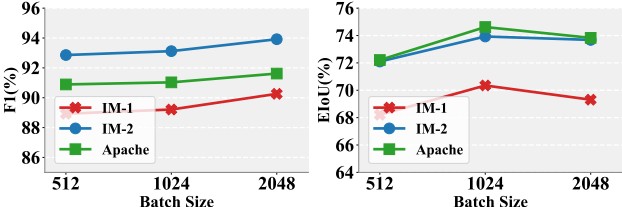

Figure 7: Ablation study on batch sizes.

```
18  bin/identity -k identity_audit
19  -a always,exit -S all -F exe=/usr/local/
20  bin/magick -k imagemagick_audit
```

IM-2:

```
1   -D
2   -b 8192
3   --backlog_wait_time 0
4   -f 1
5   -a always,exit -S fstat,getsockname,
6   connect,read,close,shutdown,sendto,
7   recvfrom,openat,writev,write,
8   bind,unlink -F exe=/usr/sbin/apache2
9   -k apache_audit
10  -a always,exit -S openat,execve,
11  read,write,close -F exe=/usr/bin/id
12  -k monitor_id
13  -w /etc/passwd -p rwxa -k passwd_audit
14  -a always,exit -S clone -F exe=/bin/sh
15  -k monitor_sh -a always,exit -S all -F
16  exe=/usr/local/bin/convert -k
17  convert_audit -a always,exit -S all -F
18  exe=/usr/local/bin/magick -k
19  imagemagick_audit
```

### E.3 Labeling Rules

We will now place the full set of labeling rules for extracting behavior units into the Table 6. We invite domain experts to define the following types of behavior unit labeling rules:

- **Event Division Rule.** Rules partition atomic application and network events within application and network logs.
- **Direct Correlation Rule.** Rules establish a direct correlation between application and network events by examining whether they possess identical key attributes.
- **Indirect Equivalence Rule.** Rules infer whether different application events represent the same execution by assessing if they are associated with identical network events.

## F More Hyper-Parameter Analysis
### F.1 Learning Rate

The results are shown in Fig. 6. It is clear that the best performance is achieved when learning rate is set to 0.001.

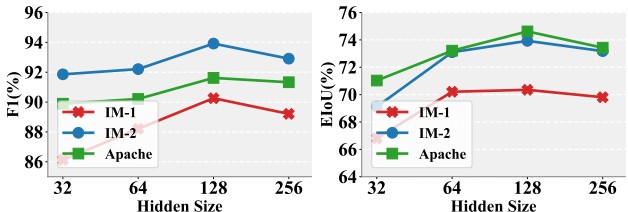

Figure 8: Ablation study on hidden sizes.

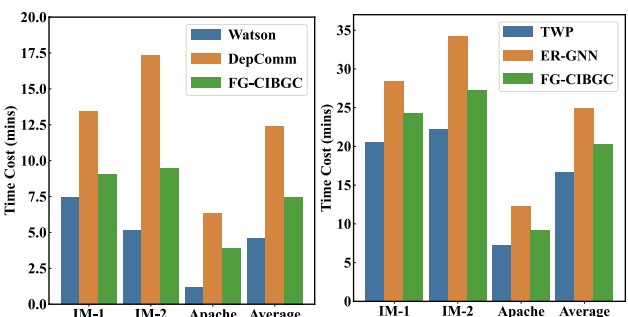

Figure 9: The time overhead compared with baselines.

### F.2 Batch Size

The results are shown in Fig. 7. It is clear that the best performance is achieved when bact size is set to 1024.

### F.3 Hidden Size

The results are shown in Fig. 8. It is clear that the best performance is achieved when hidden size is set to 128.

## G Time Analysis

In this section, we conduct an experiment to verify the time effectiveness of this framework. Specifically, as FG-CIBGC primarily comprises two stages - first partitioning the fine-grained behavior graphs, and then performing class-incremental learning - we conduct comparative evaluations against corresponding baselines in each of these stages. Results are summarized in Fig. 9.

Notably, FG-CIBGC achieves a favorable balance between time cost and performance. In terms of time efficiency of behavior graph generation, while Watson exhibits high time cost, its coarse-grained graph generation leads to inferior performance. Conversely, Dep-Comm demonstrates significantly higher time cost than FG-CIBGC, coupled with a certain degree of coarse-grained graph generation. Overall, our method effectively balances time cost and performance, exhibiting a reasonable time cost. Besides, it also should be noted

**Table 5: Overview of Dataset for FG-CIBGC Evaluation.**

| Scenarios | Vulnerability | Description |
|---|---|---|
| Apache | CVE-2021-41773 | Vulnerability allows attackers to gain control of the server and access sensitive files. |
| IM-1 | CVE-2016–3714 | The vulnerability exists because of the insufficient filtering for the file names passed to a system() call. |
| Vim | CVE-2019-12735 | Vulnerability allows remote attackers to execute arbitrary OS commands via the :source! command |
| Redis | CVE-2022-0543 | Vulnerability allows remote attackers to escape the sandbox to execute arbitrary commands. |
| Pgsql | CVE-2019-9193 | Vulnerability allows specific users to execute arbitrary code within the PostgreSQL environment. |
| ProFTPd | CVE-2019-12815 | There is a vulnerability in ProFTPD <= 1.3.6 that allows arbitrary file copying. |
| IM-2 | CVE-2022-44268 | Vulnerability leads to the reading of arbitrary files on the current operating system when converting images |
| Nginx | Path Traversal | Forgetting to include a trailing slash can result in a directory traversal vulnerability. |

**Table 6: A complete List of Labeling Rules**

| Rule Type | Rule Source | Rule Target | Fields Required | Description |
|---|---|---|---|---|
| Event Division | Network Log | Network Log | IP, Port, Time Range | In a period of time, the traffic between two (IP, Port) is considered as a connection. |
| Event Division | Apache Log | Apache Log | IP, URL | A single visit to a specific URL is considered as a separate event. |
| Event Division | PostgreSQL Log | PostgreSQL Log | IP, Port, PID | A single transaction of a database operation is considered as a separate event. |
| Event Division | Redis Log | Redis Log | IP, Port, Action | A single database operation is considered as a separate event. |
| Event Division | ImageMagick Log | ImageMagick Log | IP, Port, PID | A single operation on the image is considered as a separate event. |
| Event Division | Nginx Log | Nginx Log | IP, URL | A single visit to a specific URL is considered as a separate event. |
| Event Division | Proftpd Log | Proftpd Log | IP, Port, Filename | A single file transfer between two IP addresses is considered as a separate event. |
| Event Division | Vim Log | Vim Log | Filename, Action | A single operation on the file is considered as a separate event. |
| Direct Correlation | Network Log | Apache Log | IP, PORT,Time Range,URL | Associate logs based on the resources accessed, IP port, and time range. |
| Direct Correlation | Network Log | PostgreSQL Log | IP, PORT,Time Range,HTTP Filedata | Find database-related Network logs through HTTP file data, time, IP, and port. |
| Direct Correlation | Network Log | Redis Log | IP, PORT,Time Range,HTTP Filedata | Find database-related Network logs through HTTP file data, time, IP, and port. |
| Direct Correlation | Network Log | ImageMaick Log | IP, PORT, PID, Time Range | Find imagemagick-related Network logs through time, IP, port and PID. |
| Direct Correlation | Network Log | Nginx Log | IP, PORT,Time Range,URL | Associate logs based on the resources accessed, IP port, and time range. |
| Direct Correlation | Network Log | Proftpd Log | IP, PORT,Time Range,Filename | Associate logs based on the resources transferred, IP port, and time range. |
| Indirect Equivalence | Apache Log | PostgreSQL Log | No Field Required | PostgreSQL Log and Apache Log relate with each other by relating with any the same Network Logs. |
| Indirect Equivalence | Apache Log | ImageMagick Log | No Field Required | ImageMagick Log and Apache Log relate with each other by relating with any the same Network Logs. |

that we apply a few existing graph reduction algorithms to the generated behavior graphs. If we remove the graph reduction methods, the time costs will be significantly reduced. The results are shown in Fig. 10.

In the comparison of time cost with different backbone models, we conducted experiments using various backbones. Notably, our proposed method is able to strike a balance between time efficiency and performance.

# H  Code File

The code of our system is placed in the **code** directory of this material and detailed instructions for experiments are shown in **code/README.pdf** file.

## H.1  Python Environment Setup With Conda

Our code is written in Python3.10.8 with cuda 12.1 and pytorch 2.1.0 on Ubuntu 22.04.

Table 7: Comparison Results (Acc % and F1 %) of Class-Incremental Behavior Graph Classification Task Across Datasets. "(+)" indicates that the input to this baseline is the fine-grained behavior graphs generated by gPartition. The best results are shown in bold type and the runner-ups are underlined.

| Method | Apache | | IM-1 | | IM-2 | | Vim | | Redis | | Pgsql | | ProFTPd | | Nginx | |
| --- | --- | --- | --- | --- | --- | --- | --- | --- | --- | --- | --- | --- | --- | --- | --- | --- |
| | Acc | F1 | Acc | F1 | Acc | F1 | Acc | F1 | Acc | F1 | Acc | F1 | Acc | F1 | Acc | F1 |
| Tgminer | 68.93 | 65.15 | 69.56 | 66.33 | 62.63 | 67.67 | 69.79 | 60.13 | 74.53 | 79.06 | 62.09 | 66.43 | 69.56 | 59.75 | 63.59 | 65.91 |
| Watson | 71.54 | 69.23 | 63.59 | 56.34 | 58.43 | 66.30 | 69.49 | 56.79 | 81.69 | 82.65 | 69.43 | 61.16 | 61.93 | 52.25 | 65.84 | 54.86 |
| DepComm | 71.63 | 70.54 | 73.49 | 75.99 | 74.60 | 72.51 | 75.89 | 72.84 | 80.71 | 80.80 | 72.75 | 70.45 | 78.91 | 67.91 | 78.76 | 76.46 |
| Fine-Tuning(+) | 73.44 | 71.08 | 73.33 | 76.35 | 76.95 | 74.94 | 76.35 | 73.75 | 85.12 | 78.97 | 75.93 | 73.27 | 72.98 | 68.21 | 78.45 | 78.92 |
| EWC(+) | 76.23 | 74.95 | 74.22 | 78.02 | 80.93 | 80.74 | 83.72 | 82.42 | 84.54 | 85.15 | 89.56 | 87.32 | 87.10 | 84.34 | 83.26 | 80.84 |
| LwF(+) | 82.78 | 83.22 | 78.54 | 77.83 | 79.55 | 76.85 | 81.89 | 76.10 | 83.90 | 85.84 | 76.94 | 81.61 | 86.66 | 87.26 | 83.97 | 82.23 |
| GEM(+) | 79.34 | 76.83 | 74.28 | 74.98 | 78.39 | 76.93 | 87.46 | 82.15 | 85.06 | 84.43 | 79.45 | 81.26 | 78.12 | 81.03 | 84.57 | 83.53 |
| TWP(+) | 89.54 | 86.88 | 86.73 | 85.55 | 87.31 | 86.57 | 88.14 | 84.43 | 85.37 | 85.43 | 71.14 | 76.95 | 89.03 | 85.46 | 81.22 | 85.89 |
| CPCA(+) | 86.82 | 86.42 | 79.91 | 81.91 | 88.93 | 87.84 | 86.27 | 83.26 | 88.39 | 87.24 | 87.59 | 85.54 | 89.90 | 87.82 | 87.69 | 87.60 |
| tdGraphEmbed(+) | 77.03 | 75.12 | 66.41 | 60.78 | 68.32 | 67.37 | 78.41 | 73.81 | 79.45 | 79.03 | 73.56 | 72.62 | 73.08 | 67.63 | 65.98 | 61.54 |
| GraphERT(+) | 86.96 | 77.71 | 72.59 | 68.44 | 78.79 | 78.23 | 85.91 | 75.32 | 84.69 | 78.39 | 85.02 | 79.32 | 82.14 | 77.89 | 75.63 | 75.32 |
| TP-GNN(+) | 81.53 | 78.56 | 76.29 | 71.84 | 77.59 | 77.93 | 84.68 | 74.26 | 82.67 | 76.21 | 78.94 | 73.45 | 73.77 | 72.09 | 81.63 | 79.84 |
| **Ours** | **95.19** | **91.62** | **91.26** | **90.26** | **94.12** | **93.92** | **96.13** | **96.07** | **98.65** | **98.32** | **94.73** | **93.08** | **94.39** | **92.73** | **93.63** | **93.27** |

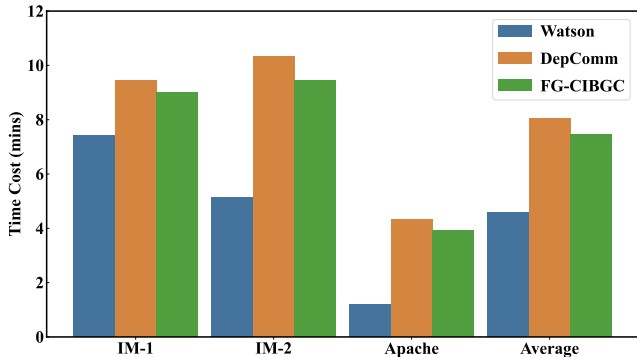

Figure 10: The time overhead comparison in terms of behavior graph partitioning without graph reduction.

install anaconda: https://repo.anaconda.com/archive/index.html.
install torch-scatter 2.1.2+pt21cu121 with the whl file downloaded from here.

```
1 $ conda create --name FG-CIBGC
2 $ conda activate FG-CIBGC
3 $ pip install -r requirments.txt
```

## H.2 Dataset

Our full dataset's compressed file size is around 2.3GB. Due to space constraints, we are only providing a sample dataset (Apache) here.

## H.3 Directory

We present a brief introduction about the directories.

## H.4 Workflow

In this section, we introduce the workflow of the overall project.

### H.4.1 Parse.
The "hlogs_parse.py" file in this directory serves as the entry point for all log preprocessing and parsing. This part is responsible for parsing audit logs, application logs, and network logs, and it generates associated JSON files for subsequent correlation with high-level behavior units and audit logs. See the following command :

```
1 $ cd Parse
2 $ python hlogs_parse.py
3 ---datasetname=$dataset
```

### H.4.2 Embedding.
The code in this directory accomplishes two main tasks. Firstly, "run.py" correlates behavior units with audit logs, ultimately generating fine-grained behavior graphs. Secondly, "run.py" executes graph embedding. See the following command :

```
1 $ cd Embedding
2 $ python run.py --dataset=$dataset
3 --kg=$algorithm
```

### H.4.3 Classification.
The code in this directory aims to produce classification results. See the following command:

```
1 $ cd Classification
2 $ python run.py --dataset=$dataset
3 -- classification=$classification
```

### H.4.4 Evaluate.
The code in this directory produces evaluation results. See the following command :

```
1 $ cd Tools
2 $ python3 evaluate.py --dataset $dataset
3 > output.txt
```

## H.5 Reproducibility

Use bash.sh to reproduce the results of performance comparison.

```
1 $ bash bash.sh
```

Use basemode_grid_search.sh to reproduce the results of grid search.

```
1    $ bash basemode_grid_search.sh
```

## I Further Explanation of Implementation

It is important to note that the labels for coarse-grained classification differ from those for fine-grained classification. When comparing classification performance, we assess the F1 score of coarse-grained classification using coarse-grained labels, while fine-grained classification is evaluated using fine-grained labels. However, when evaluating the EIoU (Edge Intersection over Union), a single coarse-grained label may correspond to multiple fine-grained labels. For example, coarse-grained label 0 corresponds to fine-grained labels 0, 1, and 2. In this case, when evaluating the EIoU, we adhere to the principle of using fine-grained labels. Consequently, the classifications made by the coarse-grained classification method for the fine-grained labels 1 and 2 are considered incorrect. We select GOOD-D [25] as the baseline OOD detector, as it is the sole open-source unsupervised graph-level OOD detection method available. We leave the exploration of alternative OOD detection techniques for future work.

## J KATE Algorithm Description

In FG-CIBGC, we choose KATE, a simple $k$NN-based sampling algorithm that does not involve much computational overhead in practice, for in-context example augmentation. Specifically, we begin by embedding all parsed log batch candidates $x_i$ from training data into vector representations $v_i$. Then, for each vectorized query $v_q$, we calculate the similarity metric $d(v_q, v_i)$ between it and all candidates, outputting the top-8 results as examples. Note that in our implementation, the similarity metric $d(v, v_i)$ represents the cosine distance as shown in Eq. 6.

$$d(v_q, v_i) := \cos(v_q, v_i) = \frac{v_q \cdot v_i}{\|v_q\|_2 \|v_i\|_2}, \tag{6}$$

Moreover, some studies have also shown that the permutation of different examples in the context can also affect the performance of ICL seriously. For example, Zhao [50] pointed out that the model's prediction for a query tends to be biased towards the closest example (i.e., recency bias), which means if the example closest to the query in the prompt is similar enough to the query, the model's prediction for the query will tend towards the results closest to the query (i.e., obtaining the correct prediction according to the nearest example's label supervision). Therefore, we choose to directly use the similarity measure $d$ obtained in the previous step to arrange these examples in ascending order, so that the example closest to the query is most similar to the query.

