# OpenReview forum: "FG-CIBGC: A Unified Framework for Fine-Grained and Class-Incremental Behavior Graph Classification"
_ACM.org/TheWebConf/2025/Conference — WWW 2025 Oral_

### Official Review · Reviewer_pXbA · 2024-11-09

**Novelty:** 4
**Technical Quality:** 4

**Review:**

This paper presents the FG-CIGBC framework tailored for Behavior Graph Classification (BGC), which comprises two main components: (1) a graph partitioning phase based on LLMs and (2) an incremental model adaptation phase. Overall, the paper effectively addresses challenges in the BGC domain and supports its claims with adequate experimental evidence.

However, from a technical standpoint, the core contributions of FG-CIGBC appear to lean more towards an application-oriented product than a novel framework. Specifically, the gPartition module utilizes LLMs for graph partitioning, while the gAdapt module employs an OOD detector in conjunction with KMeans to adapt the model to emerging, unpredictable classes in an open-world context. This approach, in my view, offers limited technical novelty.

The decision to use KMeans for class annotation is primarily motivated by its "efficiency," but it is well-known that KMeans has limitations, such as: (1) sensitivity to initial centroids, (2) sensitivity to outliers and noisy data, and (3) poor performance with unevenly distributed data. These challenges may become particularly pronounced in the gAdapt module, as it is used to annotate OOD instances. This aspect of the model requires a clearer and more detailed discussion.

Please ensure that spaces before and after figures/tables (lines 653, 774, 746, etc.) remain unchanged, as alterations are not permitted.

**Questions:**

gAdapt uses a KMeans algorithm for class annotation, despite the known limitations of KMeans, including (1) sensitivity to initial centroids, (2) sensitivity to outliers and noisy data, and (3) poor performance with unevenly distributed data. These issues are likely to be exacerbated in the context of gAdapt's use for annotating OOD instances. How does gAdapt address these challenges?

**Reviewer Confidence:**

3: The reviewer is confident but not certain that the evaluation is correct

**Scope:**

3: The work is somewhat relevant to the Web and to the track, and is of narrow interest to a sub-community

---

### Official Review · Reviewer_Wd8n · 2024-11-27

**Novelty:** 4
**Technical Quality:** 3

**Review:**

**Pros:**

1. A relatively complete framework to address the problem of finding and adapting to new classes in behavior graphs without forgetting.
2. Clear paper presentation.
3. Provide a new and comprehensive dataset to study the problem.

**Cons:**

1. A combination of LLM-based graph construction, OOD detection and incremental learning techniques, which makes the paper more focused on engineering techniques. Lack of a convincing technical contribution.
2. **Graph construction and OOD detection** or **OOD detection and incremental learning** can both lead to interesting problems, but combining all three makes the paper less focused.
3. Contribution to the incremental learning part is limited. The prototype selection method and the losses for preventing forgetting ($\mathcal{L}\_{kd}$, $\mathcal{L}\_{old}$) are not new.

**Questions:**

1. As part of the paper addressing the forgetting issue, please provide results on how well forgetting is prevented.

2. How to separately evaluate the performance of the graph partition module and the OOD detection module? Using only the final performances makes the contribution of each module unclear and reduces the soundness of the paper.

3. It is suggested that the paper could find the bottleneck in simplifying the gAdapt part without losing its effectiveness/efficiency/generalization ability, and solve that bottleneck to obtain a better technical contribution.

**Reviewer Confidence:**

4: The reviewer is certain that the evaluation is correct and very familiar with the relevant literature

**Scope:**

4: The work is relevant to the Web and to the track, and is of broad interest to the community

---

### Official Review · Reviewer_hAEB · 2024-12-01

**Novelty:** 3
**Technical Quality:** 2

**Review:**

The paper presents a unified framework termed as Fine-Grained and Class-Incremental Behavior Graph Classification (FG-CIBGC). The quality of the work is commendable, particularly in addressing an important problem in graph classification. The methodological approach appears sound. The paper is generally well-written and clear. However, certain sections, particularly those describing the technical details of the proposed methods, could be more concise and easier to follow. Providing additional diagrams or flowcharts might help clarify complex parts of the methodology.The proposed FG-CIBGC framework introduces an innovative approach by integrating fine-grained graph partitioning with class-incremental learning. The authors claim the first unified framework for fine- grained and class-incremental behavior graph classification.The problem of graph-level classification is both interesting and widely applicable. The FG-CIBGC framework has the potential to significantly impact various domains where such classification tasks are crucial. However, the text-based graph classification aspect of the method faces limitations, such as long token sequences and lack of novelty compared to other existing methods.
Pros
1 Unified Framework: Presents a cohesive framework that integrates multiple aspects of graph classification.
2 Innovative Approach: Combines fine-grained graph partitioning with class-incremental learning.
3 Wide Applicability: The proposed method has potential applications in various real-world scenarios.
Cons
1 Method Limitations: The text-based graph classification component suffers from issues such as long token sequences.
2 Novelty Concerns: While the integration is novel, some individual components may lack significant innovation compared to existing methods.

**Questions:**

1 The paper discusses the generalization ability of the proposed method, and in the experimental section, it explains how this generalization ability is demonstrated.
2 What factors can affect the performance of FG-CIBGC, and could you discuss in more detail the limitations and potential improvements of FG-CIBGC? It is recommended to add further experimental validation.
3 The overall paper lacks conciseness, and the coherence between sections is not tight enough. The length of 17 pages needs to be reduced. Besides, insufficient comparative Analysis in experiments part.
4 Based on the graph structure classification in this paper, using the promote method may bring about various issues, such as long token problems. There are many large-scale graph model structures currently available to achieve graph-level classification. Why not adopt a direct approach using large models capable of understanding graphs? The novelty is lacking in text-based large model approaches for graph classification.

**Reviewer Confidence:**

4: The reviewer is certain that the evaluation is correct and very familiar with the relevant literature

**Scope:**

3: The work is somewhat relevant to the Web and to the track, and is of narrow interest to a sub-community

---

### Official Review · Reviewer_H17m · 2024-12-03

**Novelty:** 5
**Technical Quality:** 5

**Review:**

This paper proposes FG-CIBGC, a unified framework designed to address the challenges of fine-grained and class-incremental behavior graph classification. The authors introduce a method that combines graph reduction, behavior unit generation, and model adaptation to enable effective classification of both old and new classes in an incremental manner.  The paper is well-structured and provides a clear overview of the problem, the proposed solution, and the experimental results.

The proposed approach involves several key components: graph reduction to simplify the input graphs, behavior unit generation to extract meaningful features from the logs, and model adaptation to update the classifier as new classes are introduced.  The authors demonstrate the effectiveness of their method through extensive experiments on a variety of datasets, including audit logs, net logs, app logs, and multi-source logs.

The paper provides a comprehensive analysis of the experimental results, comparing the proposed FG-CIBGC framework with baseline methods.  The results show that FG-CIBGC outperforms the baselines in both fine-grained and class-incremental settings, highlighting the robustness and adaptability of the proposed approach.

**Questions:**

1 How does the proposed FG-CIBGC framework handle the trade-off between model complexity and classification accuracy?
2 The paper mentions the use of behavior units to represent logs.  Can you provide more details on how these behavior units are generated and how they capture the essential features of the logs?
3 In the experimental results, it is noted that the proposed method outperforms baselines in both fine-grained and class-incremental settings.
 Can you discuss any potential limitations or areas for improvement in the proposed framework?

**Reviewer Confidence:**

4: The reviewer is certain that the evaluation is correct and very familiar with the relevant literature

**Scope:**

3: The work is somewhat relevant to the Web and to the track, and is of narrow interest to a sub-community

---

### Official Review · Reviewer_hDZD · 2024-12-03

**Novelty:** 6
**Technical Quality:** 7

**Review:**

Pros:
- Research tackles the critical issue of incremental and fine-grained behavior graph classification, which is quite meaningful for enhancing the accuracy and robustness of behavior analysis in web security.
- Framework effectively combines out-of-distribution detection with incremental learning to create a technical solid pipeline like lifelong learning. This integration proposes an interesting solution to the open challenge in real-world cyberattack investigation.
- Experiments are conducted in high quality across multiple datasets and baselines. Comprehensive ablation studies involving hyperparameters are also presented to provide detailed analysis into the impact of different settings on model performance.

Cons:
- Please provide citations in the introduction of Fine-Grained Emerging Behavior Graphs and Incremental Model Adaptations in L99-L129 to further support your analysis.
- It would be beneficial from careful proofreading to eliminate small errors.

**Questions:**

Please see the weaknesses above to further improve your paper.

**Reviewer Confidence:**

4: The reviewer is certain that the evaluation is correct and very familiar with the relevant literature

**Scope:**

4: The work is relevant to the Web and to the track, and is of broad interest to the community

---

### Official Review · Reviewer_PvpX · 2024-12-03

**Novelty:** 7
**Technical Quality:** 5

**Review:**

Pros:
1. Innovative task: The paper proposes a novel work on unifying fine-grained classification and class-incremental learning for behavior graph classification, which I have never heard of.
2. Novel methodology: The introduction of LLM under the in-context learning paradigm demonstrates an interesting approach to to process multi-source logs, which enhances the ability to identify semantically similar logs.
3. Empirical validation: Extensive experiments demonstrate the framework's superiority over state-of-the-art methods, providing strong evidence of its effectiveness. The introduction of EIoU provides a reasonable insight for behavior graph classification.
4. Smooth writing: The whole paper is well-organized and easy to follow, featured with well-designed illustrations to introduce the new task and the method.

Cons:
While the originality and practical significance of this work make it a valuable contribution to web graph learning community, there are still minor errors that should be corrected. E.g., “Six class-incremental incremental learning methods:” in L663.

**Questions:**

The introduction of the EIoU metric is a significant contribution. Can you provide some examples on how EIoU compares to traditional metrics like F1-Score in terms of sensitivity to fine-grained classification tasks?

**Reviewer Confidence:**

4: The reviewer is certain that the evaluation is correct and very familiar with the relevant literature

**Scope:**

3: The work is somewhat relevant to the Web and to the track, and is of narrow interest to a sub-community

---

### Official Review · Reviewer_NtLm · 2024-12-04

**Novelty:** 6
**Technical Quality:** 6

**Review:**

This paper tackles the novel and challenging task of Fine-Grained and Class-Incremental Behavior Graph Classification (FG-CIBGC). The proposed framework addresses two critical challenges: (i) mining fine-grained semantics in multi-source logs using Large Language Models (LLMs) under the In-Context Learning (ICL) paradigm and (ii) bridging the gap between Out-of-Distribution (OOD) detection and class-incremental learning. The framework integrates two key modules: gPartition for fine-grained behavior graph generation and gAdapt for incremental model adaptation. The authors further provide a benchmark dataset and a novel metric, Edge Intersection over Union (EIoU), for evaluating the proposed method.

## Strengths
- The authors address a significant gap in Behavior Graph Classification by introducing the FG-CIBGC framework, which is of importance. This work stands outr as the first to explore fine-grained partitioning and class-incremental adaptation simultaneously in the graph domain.
- The proposed modules, gPartition and gAdapt, are designed logically and demonstrate well-founded integration of LLM-based log processing and advanced OOD detection. Integrating foundation models to guide the problem-solving of the specific problem.
- Extensive experiments validate the superiority of FG-CIBGC over existing baselines, achieving consistent improvements in metrics such as EIoU and F1-score, with a robust benchmark with diverse log sources, realistic attack scenarios, and a fine-grained dataset
- The link for code and datasets validate the reproducibility.

**Questions:**

## Weakness, typos, and questions
- While the paper is generally well-written, minor typographical errors (e.g., "operatons" instead of "operations") need to be solved.
- How computationally expensive was the use of In-Context Learning (ICL) during the training and inference stages? Did the authors consider any strategies to optimize the cost of ICL, such as prompt engineering
- Have the authors evaluated the performance of FG-CIBGC using other LLMs, both open-source (e.g., LLaMA, BLOOM) and closed-source (e.g., GPT-4)? Were there significant differences in the performance of gPartition across these models?
- Could the proposed FG-CIBGC framework generalize to other graph-based tasks beyond behavior graph classification? For instance, how well might it adapt to applications like protein structure classification, 3D mesh segmentation, or social network analysis?
- Could the authors conduct additional experiments varying the ratio of unknown to known classes (e.g., 7:3, 5:5) to assess the robustness of FG-CIBGC in different real-world scenarios?

**Reviewer Confidence:**

3: The reviewer is confident but not certain that the evaluation is correct

**Scope:**

4: The work is relevant to the Web and to the track, and is of broad interest to the community